# Liquid Madelung energy accounts for the huge potential shift in electrochemical systems

Norio Takenaka [1], Seongjae Ko[1], Atsushi Kitada [1] & Atsuo Yamada [1,2] ✉

Achievement of carbon neutrality requires the development of electrochemical technologies suitable for practical energy storage and conversion. In any electrochemical system, electrode potential is the central variable that regulates the driving force of redox reactions. However, quantitative understanding of the electrolyte dependence has been limited to the classic Debye-Hückel theory that approximates the Coulombic interactions in the electrolyte under the dilute limit conditions. Therefore, accurate expression of electrode potential for practical electrochemical systems has been a holy grail of electrochemistry research for over a century. Here we show that the 'liquid Madelung potential' based on the conventional explicit treatment of solid-state Coulombic interactions enables quantitatively accurate expression of the electrode potential, with the Madelung shift obtained from molecular dynamics reproducing a hitherto-unexplained huge experimental shift for the lithium metal electrode. Thus, a long-awaited method for the description of the electrode potential in any electrochemical system is now available.

The achievement of a carbon-neutral society is imperative for addressing the current environmental crisis and securing sustainable development for future generations. Owing to the exceptional dedication of numerous researchers, significant advances have been achieved in the development of energy storage and conversion technologies such as batteries, fuel cells, and water electrolysis, enabling the widespread development and use of electric and fuel cell vehicles that do not rely on petrol, as well as the production of clean hydrogen using renewable energy sources. For significant progress in overcoming the energy and environmental issues arising from fossil fuel consumption, it is necessary to maximize the performance of these advanced electrochemical systems based on sophisticated design concepts in order to enable their practical use.

The electrode potential $E$ is undoubtedly the most fundamental and important concept that regulates any electrochemical reaction. Many of the recent studies have revealed that long-lasting challenges in next-generation batteries can be readily addressed through the electrolyte design based on the strategic control of the electrode potential. For instance, a highly reversible Li plating/stripping was achieved by increasing the electrode potential of Li metal ($E_{Li/Li+}$), thereby reducing the thermodynamic driving force for electrolyte reduction[1]. Furthermore, exceptionally stable $SiO_x$/$LiNi_{0.5}Mn_{1.5}O_4$ batteries have been developed by optimizing the overall potential diagram of the electrodes and electrolytes[2]. Therefore, the quantitative description of $E$ is of vital importance and has been a holy grail of electrochemical research for over 100 years since the introduction of the Debye-Hückel theory in 1923[3] that is valid only for electrolytes in the limit of infinite dilution as described below. Tremendous efforts have been devoted to extend the Debye-Hückel theory to practical concentration region, however, explicit treatment of the long-range Coulombic interaction in highly fluctuating many-body liquid system has long been unrealistic[4,5], until the recent advances and maturity in molecular dynamics (MD) simulations. Currently, there is no suitable physical model for quantitatively accurate theoretical expression of the electrode potential in the concentrated regime relevant for most electrolytes used for practical energy applications.

[1]Department of Chemical System Engineering, School of Engineering, The University of Tokyo, Tokyo, Japan. [2]Sungkyunkwan University Institute of Energy Science & Technology (SIEST), Sungkyunkwan University, Suwon, Korea. ✉e-mail: yamada@chemsys.t.u-tokyo.ac.jp

## Results and discussion

### Anomalous potential shift beyond Debye-Hückel theory

Generally, the electrode potential is known to be dependent on the activity of the ions in the electrolyte[1,2,6–13]. For instance, the thermodynamic redox potential of Li metal (Li/Li$^+$) shifts strongly (surprisingly by more than 0.6 V) depending on the electrolyte salt concentration in the lithium bis(fluorosulfonyl)imide/propylene carbonate (LiFSI/PC) system (Fig. 1). According to the Nernst equation, the potential shift of Li/Li$^+$ ($\Delta E_{Li/Li+}$) is described using the activity of Li$^+$ ($a_{Li+}$) with reference to 1 mol L$^{-1}$ (for which the activity can be assumed to be unity):

$$\Delta E_{Li/Li+} = \frac{RT}{F}\ln a_{Li+} = \frac{RT}{F}\ln[\gamma_{Li+}m_{Li+}] \qquad (1)$$

where $R$, $T$, $F$, $\gamma_{Li+}$, and $m_{Li+}$ are the gas constant, temperature, Faraday constant, activity coefficient, and molality of Li$^+$, respectively. The experimental potential upshift deviates markedly from the ideal curve for the concentrations above 1 mol L$^{-1}$ (0.9 mol kg$^{-1}$), giving rise to a huge gap of over 0.3 V at the highest concentration that corresponds to a huge activity coefficient as large as $\gamma_{Li+} = 10^5$. This cannot be explained by the classical Debye-Hückel theory[3], in which the Coulombic interactions in the electrolyte are approximated in the infinitely dilute regime. Although phenomenological extensions of the Debye-Hückel theory to higher concentrations through the inclusion of complex empirical calibration terms with adjustable parameters may be partially applicable for an apparent fitting to the experimental data[4,5], the physical meaning and key mechanisms governing the potential shift remain unclear. Solving this fundamental problem requires explicit treatment of the Coulombic interaction in an electrolyte in the higher-concentration region, leading us to adopt the concept of 'liquid Madelung potential' ($E_{LM}$) based on the conventional Madelung energy concept used in solid-state science as explained below.

Experimentally, a huge potential shift cannot be observed in a typical battery system (two-electrode cell) and may have been overlooked in many cases, because identical potential shifts occur simultaneously at both the cathode and the anode as a result of the common changes in the intrinsic energetics of the electrolyte, $\Delta E = (RT/F)\ln a_{Li+}$[13]. For instance, the intercalation/deintercalation potentials of Li$^+$ into/from Li$_{4+x}$Ti$_5$O$_{12}$ (LTO, $x$ ~ 2) and Li$_y$FePO$_4$ (LFP, $y$ ~ 0.3) shift by an amount identical to that of the shift in the redox potential of Li/Li$^+$ (Fig. 2); here, the IUPAC recommended internal reference, Fe$^{3+}$/Fe$^{2+}$ in ferrocene, was used to monitor each of the relative potential differences independently. Details of the methodology are given in the methods section. The identical potential upshifts also indicate that a passivation film on the electrodes has no impact on $E$[1].

To confirm that this trend is general, three types of FSI-based electrolytes with different solvents were examined. Figure 3a–c shows potential shifts $\Delta E_{Li/Li+}$ measured for LiFSI/ethylene carbonate (LiFSI/EC), LiFSI/PC and LiFSI/sulfolane (LiFSI/SL) with various $m_{Li+}$. The experimental potential shift $\Delta E_{Li/Li+}$ (black circles) at each concentration is displayed with respect to 1 mol L$^{-1}$ ($m_{Li+} = 0.8$ mol kg$^{-1}$ for LiFSI/EC and 0.9 mol kg$^{-1}$ for LiFSI/PC and LiFSI/SL). Notably, the plots of the experimental values of $\Delta E_{Li/Li+}$ are consistent with the logarithmic relationship (black solid lines). However, the experimental values deviate markedly from those obtained assuming $\gamma_{Li+} = 1$ in the Nernst equation (black dashed lines). Moreover, the observed large upshift (0.35 V) at the highest concentration (7.6 mol kg$^{-1}$) in the LiFSI/EC system can be converted to a value as large as $\gamma_{Li+} = 10^5$, which is definitely beyond the scope of the Debye-Hückel theory.

### Liquid Madelung potential

Having confirmed the general trend, we sought to understand the physics underlying this anomalous behavior. To reveal the mechanism governing the large potential upshift, we focused on the explicit calculation of the electrostatic potential at the Li$^+$ sites, which cannot be treated by the conventional Debye-Hückel theory. To quantitatively relate the local coordination structure and thermodynamics of the system, we introduce the liquid Madelung (Ewald) potential ($E_{LM}$) of Li$^+$, which is obtained by summing all of the electrostatic interactions of the surrounding solvents/ions for each atom individually:

$$E_{LM} = \frac{1}{N_{Li+}}\left\langle\sum_i \frac{q_i}{4\pi\varepsilon_0 r_i}\right\rangle \qquad (2)$$

where $N_{Li+}$, $\varepsilon_0$, $q$, and $r$ are the number of Li$^+$ ions, the permittivity of vacuum, atomic charge of the molecules/ions, and the distance $r$ of the atoms of the surrounding solvents/ions from the central Li$^+$, respectively. The bracket < > denotes time averaging in MD simulations. Note that $E_{LM}$ is calculated based on the particle mesh Ewald method (see S1 in the Supplementary Information for more details)[14]. Details of the MD simulations are described in the methods section.

A conceptual derivation of the liquid Madelung (Ewald) potential $E_{LM}$ is shown in Fig. 4. An analogous concept is the classical Madelung potential for inorganic crystals (Fig. 4a)[15], which is the sum of electrostatic interactions of the ions approximated by point charges fixed at a crystal lattice, yielding discrete changes in the Coulombic energy as a function of $r$ (distance from the central Li$^+$). To calculate $E_{LM}$, all of the atoms of the solvents/ions are approximated by the point charges determined by the restrained electrostatic potential (RESP) method (Fig. S1)[16]. However, ions and solvent molecules in electrolyte solutions are highly mobile and show a certain randomness in their relative positions, in contrast to the ions in inorganic crystals. Therefore, $E_{LM}$ is evaluated by space-averaging the $E_{LM}$ of each Li$^+$ over all Li$^+$ ions in a calculation cell and subsequently time-averaging $E_{LM}$ for all Li$^+$ across all snapshots. Consequently, the spatiotemporal average results in a smooth curve of Coulombic energy vs. $r$ (Fig. 4b, c), in contrast to the discontinuous variation of the Coulombic energy with $r$ for each Li$^+$ in a snapshot at a certain time. Sample calculations of the Coulombic energy vs. $r$ for the LiFSI/EC system are shown in Fig. S2.

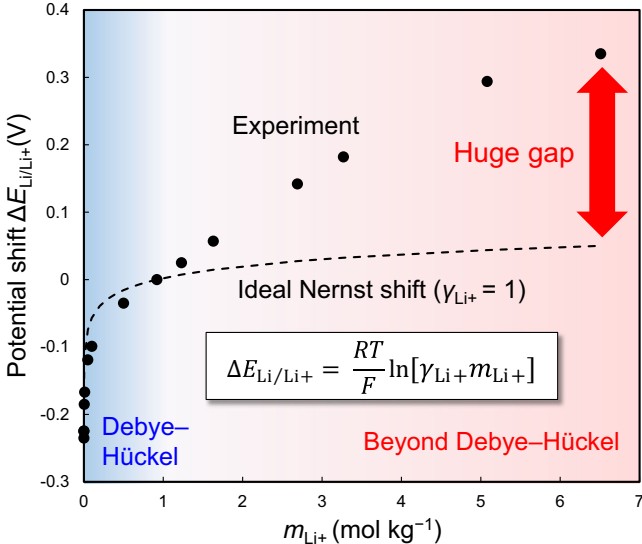

**Fig. 1 | Anomalous potential shift beyond Debye-Hückel theory.** The experimental shift of redox potentials of Li metal ($\Delta E_{Li/Li+}$) in LiFSI/PC electrolytes as a function of salt concentration $m_{Li+}$ with reference to 1 mol L$^{-1}$ or 0.9 mol kg$^{-1}$. Black dashed lines represent the potential shift based on the ideal Nernst equation, wherein the activity coefficient $\gamma_{Li+}$ is assumed to be unity, showing significant deviation from the experimental values at high concentrations. The huge gap can be explained neither by the classical Debye-Hückel theory nor by its phenomenological extensions.

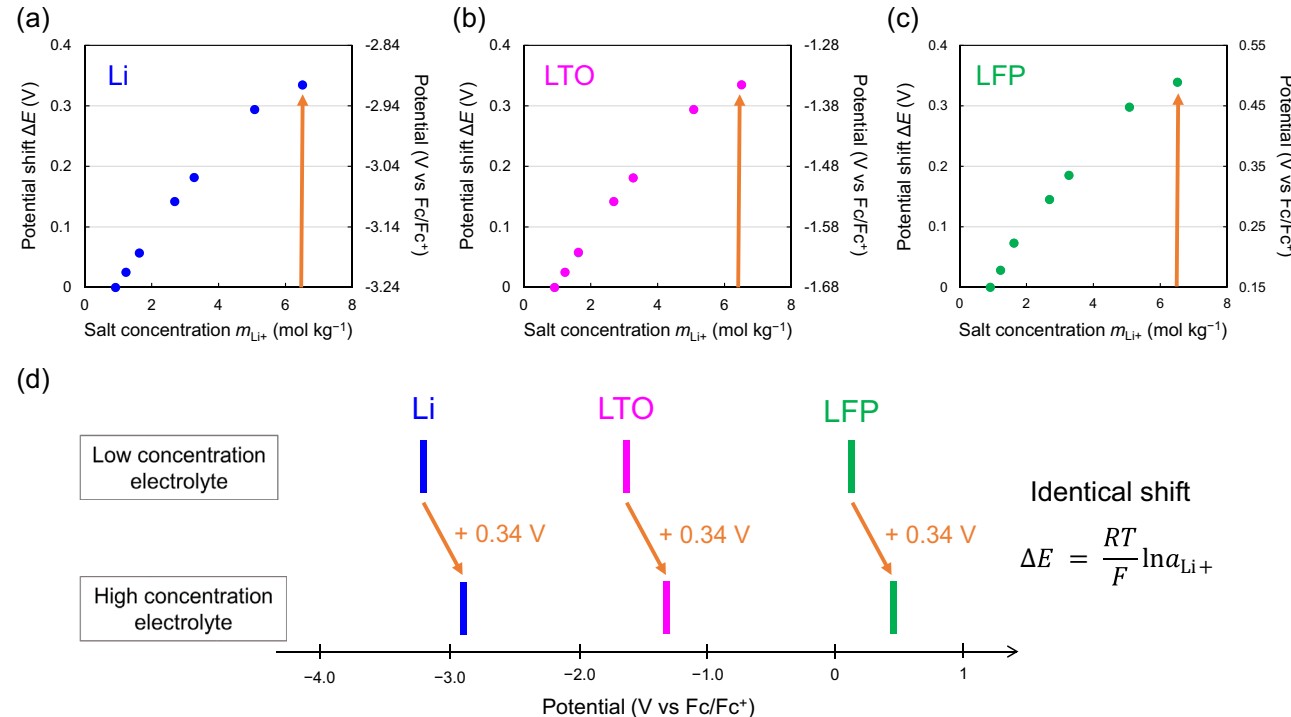

**Fig. 2 | Identical potential shifts independent of the electrode in the common electrolyte. a–c** The experimental potential shifts of **a** Li, **b** Li$_{4+x}$Ti$_5$O$_{12}$ (LTO, $x \sim 2$) and **c** Li$_y$FePO$_4$ (LFP, $y \sim 0.3$) in LiFSI/PC electrolytes as a function of salt concentration ($m_{Li+}$) with reference to 1 M or 0.9 mol kg$^{-1}$. **d** Potential diagram demonstrating the identical potential upshifts for Li, LTO, and LFP upon increasing salt concentration.

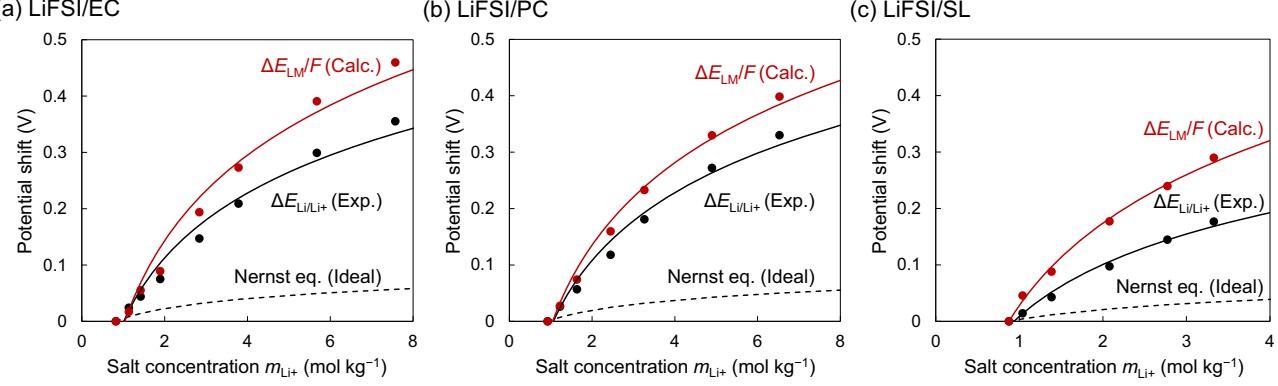

**Fig. 3 | Liquid Madelung potential explains the upshift of electrode potential of Li ($E_{Li/Li+}$).** The experimental shifts of $E_{Li/Li+}$ (black circles, $\Delta E_{Li/Li+}$) and the calculated potential shifts of the liquid Madelung (Ewald) potential (red circles, $\Delta E_{LM}/F$; see Fig. 4 and text for details) were designated as the shift from the data for the lowest concentrations and plotted as a function of salt concentration $m_{Li+}$ for **a** LiFSI/EC, **b** LiFSI/PC, and **c** LiFSI/SL electrolytes. $\Delta E_{Li/Li+}$ and $\Delta E_{LM}/F$ exhibit a logarithmic relationship (black and red solid lines). The black dashed lines represent the potential shift based on the ideal Nernst equation, where the activity coefficient $\gamma_{Li+}$ is assumed to be unity.

Applying the aforementioned protocols based on MD simulations, we accurately and directly calculated, for the first time[17,18], the 'Madelung potential' of Li$^+$-containing electrolyte solutions by summing the electrostatic potentials of the surrounding atoms. The variations in the shift of the calculated $E_{LM}$ (i.e., $\Delta E_{LM}/F$, divided by the Faraday constant for unit conversion from eV to V) versus $m_{Li+}$ are presented in Fig. 3a–c (red circles) and reasonably reproduce the concentration dependence of the experimental upshift $\Delta E_{Li/Li+}$ (black circles). Such strong correlations between $\Delta E_{LM}/F$ and $\Delta E_{Li/Li+}$, which are now confirmed to be general for the LiFSI solutions of EC, PC and SL, indicate the predominant contribution of the liquid Madelung potential to the observed potential upshift. Noteworthy is that the concept of $E_{LM}$ is valid for other alkali metals (to be reported elsewhere) as well as for

multivalent metal ions, as typically demonstrated for zinc bis(trifluoromethanesulfonyl)imide (Zn(TFSI)$_2$)/PC electrolytes in Fig. S3. To emphasize, the liquid Madelung potential has never been calculated directly according to Eq. [2], though the term was used for conceptual interpretation of the experimental spectra measured for ionic liquids[17–20].

**Mechanism of potential upshift**

Since the liquid Madelung (Ewald) potential is closely related to the local coordination environment, we analyzed the Raman spectra of LiFSI/EC (Fig. 5a and b), LiFSI/PC (Fig. S4a and b), and LiFSI/SL (Fig. S4c and d). With an increase in the salt concentration, Li$^+$ is predominantly coordinated by FSI$^-$ anions rather than by solvents, and

(a) Classical Madelung potential for inorganic crystals

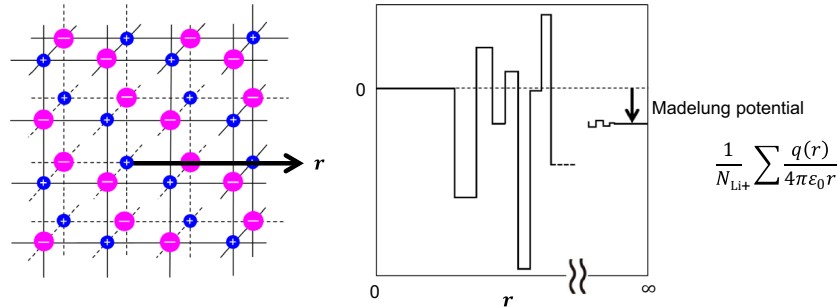

(b) Liquid Madelung potential ($E_{LM}$) for low concentration electrolyte solutions

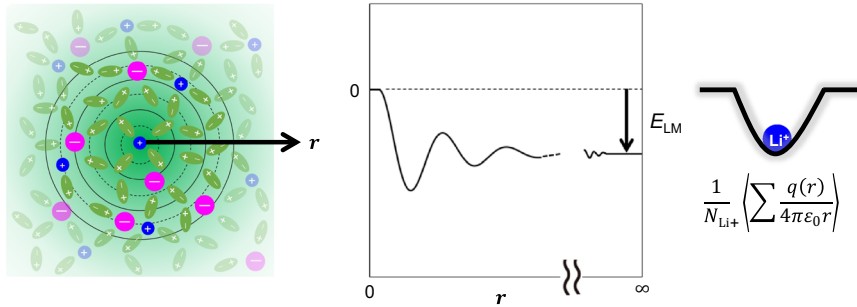

(c) $E_{LM}$ for high concentration electrolyte solutions

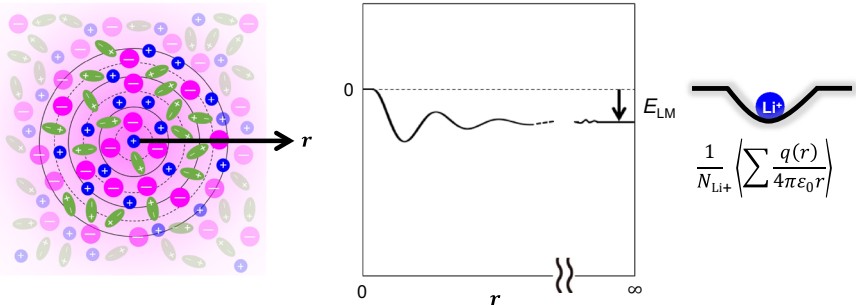

**Fig. 4 | Concept of liquid Madelung (Ewald) potential ($E_{LM}$) in electrolytes.** $E_{LM}$ represents the sum of the electrostatic interactions of the surrounding ions and/or solvents with the central Li$^+$. For inorganic crystals, this concept is known as the classical Madelung potential (a), where ions are fixed in a crystal lattice, yielding the discrete changes in the Coulombic energy vs. position $r$. However, since solvent molecules and ions are mobile in electrolyte solutions, $E_{LM}$ is calculated by both time- and space-averaging the potentials from the snapshots obtained by molecular dynamics simulations, resulting in a smooth variation of Coulombic energy with $r$ (b). Li$^+$ site potential was significantly shallower at higher salt-concentrations (c), leading to a larger potential upshift ($\Delta E_{LM} > 0$; Fig. 3a–c).

the dominant coordination structure changes from a solvent-separated ion pair (SSIP; Li$^+$(solvent)$_4$ and free FSI$^-$) to a contact ion pair (CIP) and/or cation-anion aggregates (AGG)[21]. MD simulations also support this trend (Fig. 5c and d). The calculated radial distribution functions (RDFs; g($r$)) show that the peak intensity is considerably larger for Li-O$_{EC}$ than for Li-O$_{FSI^-}$ at low concentrations (Fig. 5e), whereas the opposite trend is observed for highly concentrated media (Fig. 5f). In addition, the coordination numbers (CNs) for EC and FSI$^-$ obtained from the number distribution functions $n(r)$ indicate that the dominant coordination state is SSIP at low concentrations and AGG at high concentrations (Fig. S5); this is also consistent with the Raman spectra. Furthermore, similar results were obtained for the PC- and SL-based electrolytes (Fig. S6). Consequently, it is verified that MD simulations provide a good description of the actual coordination environment, indicating that the calculated values of $E_{LM}$ are reasonable.

The variation in $E_{LM}$ of Li$^+$ with the changes in the coordination environment can be explained based on the electron density distributions[22] for each solvent/anion. As displayed in Fig. 5g, electrons are strongly localized on oxygen atoms in EC, whereas they are delocalized in FSI$^-$ over a considerably larger region. Such electron-localized oxygen atoms also exist in PC and SL (Fig. S7). This, in turn, distinguishes the depth of the Li$^+$ site potentials. Considering the Coulombic energy gain due to the higher electron density in the first-coordination sphere, Li$^+$ is electrostatically more stable when Li$^+$ is strongly solvated by electron-localized oxygen atoms in the EC solvents, whereas it is unstable when coordinated by the electron-delocalized FSI$^-$ (Fig. 5g). Therefore, upon a change in the dominant first-coordination species from the electron-localized solvent to the electron-delocalized anion (e.g., from SSIP to CIP/AGG with increasing the salt concentration), $E_{LM}$ of Li$^+$ becomes significantly shallower, as described by eq. [2]; this causes an upshift in the potential.

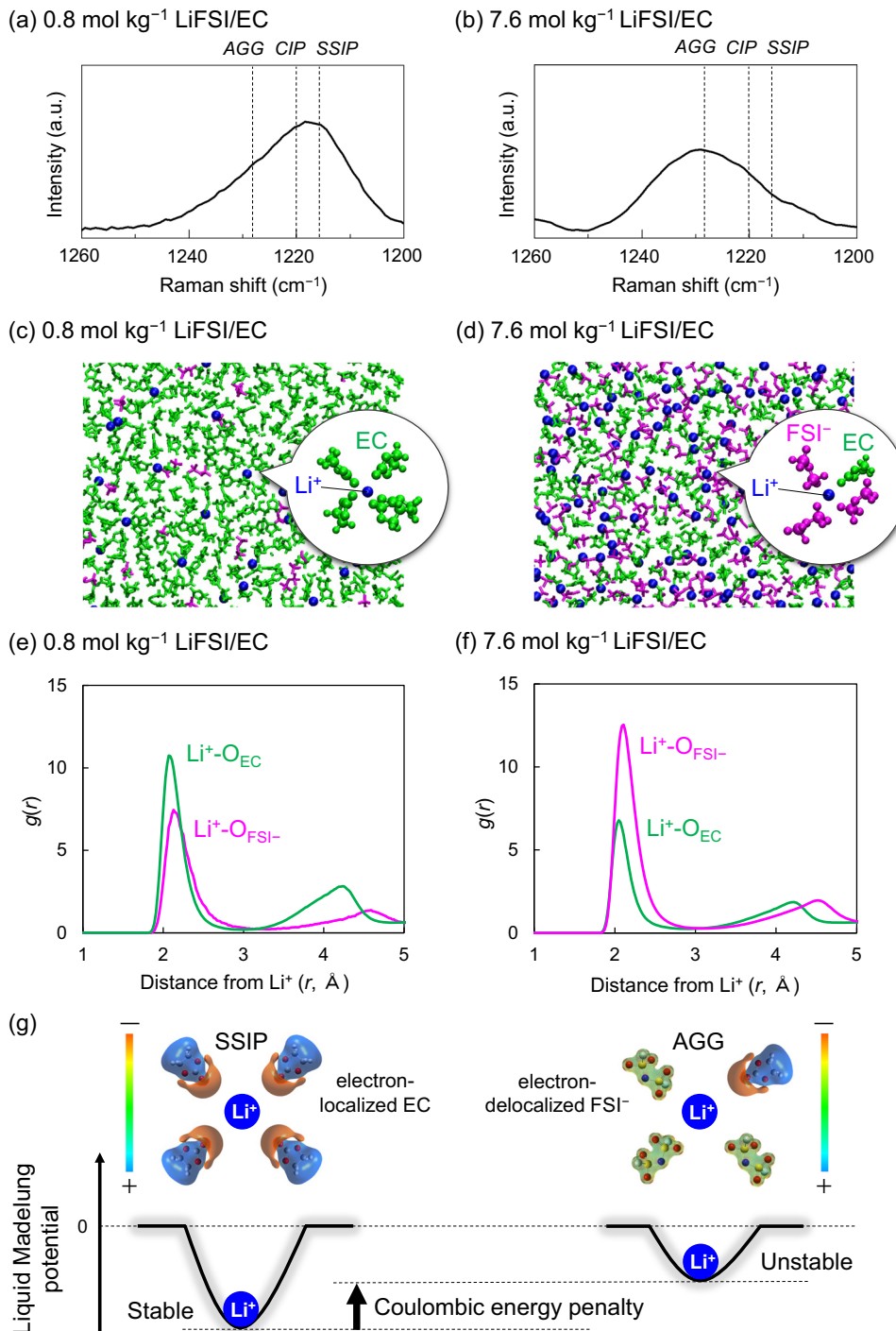

**Fig. 5 | Comparative scheme of Li+ site potential (liquid Madelung potential, $E_{LM}$) in 0.8 mol kg⁻¹ and 7.6 mol g⁻¹ LiFSI/EC electrolytes.** (a) (b) Raman spectra from which the coordination state of FSI⁻ (solvent-separated ion pair (SSIP), contact ion pair (CIP), cation-anion aggregates (AGG); black dashed lines) can be evaluated. (c) (d) Typical snapshots obtained from molecular dynamics simulations with magnified images of first-coordination-shell environment around Li⁺. (e) (f) Calculated radial distribution functions $g(r)$ (solid lines) of EC (green) and FSI⁻ (magenta). (g) Mechanism of potential upshift: Coulombic energy penalty caused by a change in the dominant local coordination from electron-localized EC to electron-delocalized FSI⁻. It should be noted that $E_{LM}$ is a converged value that is obtained by taking into account the electrostatic interactions of all constituent atoms up to infinite distances.

Note that $E_{LM}$ is a converged value that is obtained by taking into account the Coulombic interactions of all constituent atoms up to infinite distances.

As a general thermodynamic expression, the partial molar Gibbs energy or chemical potential of Li⁺ in electrolyte can be converted to the Li electrode potential as follows:

$$\frac{dG}{dn_{Li^+}} \equiv \mu_{Li^+} = FE_{Li/Li^+} \tag{3}$$

where $n_{Li^+}$ is the number of $Li^+$ ions. Since $E_{LM}$ contributes to the enthalpy term, the difference between $\Delta E_{Li/Li^+}$ and $\Delta E_{LM}$ may originate in part from the entropy term which cannot be calculated and is thus neglected in this case. Nonetheless, as shown in Fig. 3a–c, the $\Delta E_{LM}/F$ vs. concentration plots indicate a logarithmic relationship with salt concentration and reproduce $\Delta E_{Li/Li^+}$ well; this strongly indicates that $\Delta E_{LM}$ is the dominant contribution to the Gibbs energy whereas the entropy contribution is small for the systems considered in this work (see S2 in the Supplementary Information for more details).

Furthermore, the concept of upshifting $E_{Li/Li^+}$ by ligand substitution from electron-localized solvents to electron-delocalized anions can be extended to the locally concentrated case; for instance, in $1.5\,mol\,L^{-1}$ LiFSI in tetrahydrofuran/toluene (THF/toluene). Upon replacing THF with a non-polar solvent (toluene) to dilute the mixture, the AGG structure becomes more dominant (see Fig. S8 in the Supplementary Information)[23], upshifting $E_{Li/Li^+}$ (Fig. S9). Moreover, the deviation of $\Delta E_{Li/Li^+}$ from $\Delta E_{LM}/F$ is quite small, implying that the entropy change upon dilution is also small; this is presumably because the entropy term, which is governed by the number of the possible configurations of ions and solvent molecules, does not change considerably upon THF/toluene replacement. As a result of the fixed concentration of $1.5\,mol\,L^{-1}$, the upshift in $E_{Li/Li^+}$ and $E_{LM}$ is better explained by the Coulombic energy penalty alone, with only minimal contribution from the configurational entropy.

Here we show that the liquid Madelung potential, which explicitly treats Coulombic interactions in practically relevant electrolyte solutions, enables a quantitative physical interpretation of the huge potential shift beyond Debye-Hückel theory (1923). Through an exploration of interdisciplinary across solid-state and liquid-state materials science with an eye to their common characteristics as condensed systems, a microscopic and rigorous description for the electrode potential that does not rely on phenomenological derivation of the activity coefficient parameters is established. MD simulations play a vital role in determining the solution structure and calculating the total Coulombic interactions in the liquid phase. This simple but hitherto-overlooked approach can be applied to all type of electrochemical reactions and will contribute to further advances in the development of sophisticated industrial electrochemical systems in applications such as batteries, electrolysis, electroplating, and smelting.

## Methods
### Experimental study
The electrolytes were prepared by adding LiFSI (Nippon Shokubai) into several solvents in an Ar-filled glove box. Raman spectroscopy measurements were performed using an NRS-5100 spectrometer (JASCO, 532 nm laser). The samples were sealed in a quartz tube in an Ar-filled glove box to prevent air contamination. The Raman spectra were calibrated using a Si plate ($520.7\,cm^{-1}$). To estimate the potential shift of given electrodes (Li, LTO, and LFP), cyclic voltammetry was performed using a VMP3 potentiostat (BioLogic) in a three-electrode cell (Pt as the working electrode, and given electrodes (Li or LTO or LFP) as the counter and reference electrodes), with various electrolytes containing $1\,mmol\,L^{-1}$ ferrocene (Fc, Sigma Aldrich) at 298 K[24–26]. Similarly, the potential shift of $Zn/Zn^{2+}$ in various electrolytes ($Zn(TFSI)_2$ with a PC solvent) was determined using a three-electrode cell (Pt as the working electrode and Zn metal as the counter and reference electrodes) at 298 K. The cyclic voltammetry was conducted at a scan rate of $<5\,mV\,s^{-1}$ in an Ar-filled glove box, with cut-off conditions set at 2.8 to 3.6 V (for Li), 1.0 to 2.0 V (for LTO), −0.8 to 0.2 V (for LFP), and 0.25 to 1.05 V (for Zn), respectively.

The lithiated $Li_{4+x}Ti_5O_{12}$ electrode ($x \sim 2$) was prepared as follows. First, the 2032-type coin half-cell was fabricated with Li metal (Honjo Metal, 50 µm thickness, diameter of 1.2 cm) and an electrode composed of 80 wt% of $Li_4Ti_5O_{12}$ powder, 10 wt% of carbon additives (Li400, Denka), and 10 wt% of poly vinylidene difluoride (PVDF, Kureha) binder. Then, the pre-lithiation process was conducted in the commercial $1.0\,mol\,L^{-1}$ $LiPF_6$/EC:DMC (1:1, v-v) electrolyte (Kishida) at a constant current of 0.2 C-rate for 3 h at 298 K using a charge-discharge machine (TOSCAT). The lithiated $Li_{4+x}Ti_5O_{12}$ electrode was carefully removed from the coin cell and washed with 1,2-dimethoxyethane (DME) several times in the glove box. The delithiated $Li_yFePO_4$ ($y \sim 0.3$) was prepared outside of the glove box by chemically oxidizing $LiFePO_4$ powder in $NO_2BF_4$ solution in acetonitrile. The 80 wt% of delithiated $Li_yFePO_4$ ($y \sim 0.3$) powder, 10 wt% of carbon additives, and 10 wt% of PVDF binder were mixed well in the N-methylpyrrolidone (NMP, Wako) solvent. The obtained slurry was coated onto an Al current collector and dried in an oven at 80 °C under vacuum. The 2032-type coin-cell parts (Hohsen), Li metal, and Zn metal (Nilaco, 50 µm thickness, diameter of 1.2 cm) were used as received. A glass-fiber separator (GC50 Advantec) was used to ensure the immersion of all electrolytes. To ensure the proper functioning of counter and reference electrodes, a high mass loading ($\sim 5\,mg\,cm^{-2}$) of lithiated $Li_{4+x}Ti_5O_{12}$ and delithiated $Li_yFePO_4$ electrodes was applied to the cyclic voltammetry test.

### Computational study
MD simulations were conducted in the given electrolytes using the Amber 16 software package. Details of the calculation models are presented in Tables S1 and S2, where the numbers of solvent molecules and ions correspond to the experimental composition. In the simulations, the generalized AMBER force field[27] was used for all chemical species. The atomic point charges were obtained via the RESP method with density functional theory calculations at the B3LYP/cc-pvdz level, using the Gaussian 16 software package (Fig. S1). The time step was set to 1 fs using the SHAKE method[28] that constrains the bond distances between hydrogen atoms and heavy atoms. The sizes of the simulation cells were allowed to adjust by carrying out NPT-MD simulations (1 bar and 298 K). Then, using NVT-MD simulations (298 K), the systems were equilibrated for 1 ns, followed by 10 ns production runs. The ionic charges are scaled by a factor of 0.8 based on the previous studies of concentrated (or ionic liquid) electrolytes[29–33]. The calculated solution structures (Figs. 5e, f, S6, S8c, and d) reproduced the experimental Raman spectroscopy data well (Figs. 5a, b, S4, S8a, and b). The molecular electrostatic potentials (Figs. 5g and S7) were plotted using the GaussView 6.0 software. The images were obtained by mapping the electrostatic equipotential surface onto the electron density (isovalue = 0.05 e $Å^{-1}$). Moreover, $E_{LM}$ was evaluated by averaging the sum of all of the electrostatic interactions from the constituent atoms in surrounding electrolyte solvent molecules and ions to each $Li^+$ using the particle mesh Ewald method. The potentials were compared using different calculation levels, HF and M06-2x, where all methods show reasonable agreement with the experimental potential shift (Fig. S10).

## Data availability
All the relevant data are included in the paper and its Supplementary Information. Source data are provided with this paper.

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

## Acknowledgements

This work was supported by the Ministry of Education, Culture, Sports, Science, and Technology (MEXT) Programme: Data Creation and Utilization Type Materials Research and Development Project (A.Y.; Grant Number JPMXP1121467561), Japan Society for the Promotion of Science (JSPS) Grant-in Aid for Scientific Research (S) (A.Y.; Grant Number 20H05673).

## Author contributions

A.Y. proposed the concept and directed the project. N.T. performed computational simulations and analyzed the data. S.K. conducted experiments. N.T., S.K., A.K., and A.Y. wrote the manuscript.

## Competing interests

The authors declare no competing interests.
