## [Peer Review File · Nature Communications]

Liquid Madelung energy accounts for the huge potential shift in electrochemical systemsREVIEWER COMMENTS

Reviewer #1 (Remarks to the Author):

In this work, the authors reported that the “liquid Madelung potential” (ELM) based on the conventional explicit treatment of solid-state Coulombic interactions enables the quantitatively accurate expression of the electrode potential. Based on the data presented, I think the manuscript can satisfy the requirement for Nature Communications. Issues are addressed below.

Comment 1: The author has well introduced the explicit method to interpret the huge potential shift beyond Debye-Hückel's theory. In this work, Li^+ is fully discussed, while I'm curious if this method can also extend to other multivalent metal-ion batteries. If the solvation structure is unclear or complicated if this method still works.

Comment 2: In terms of accuracy, the author mentioned “excellent quantitative accuracy”. A detailed discussion regarding accuracy should be addressed. If there is any assumption before using this method. What are these conditions?

Comment 3: In this work a high concentration was used as an experimental result, does this also work for super high concentrated electrolytes? Such as water-in-salt electrolytes or deep eutectic electrolytes.

Comment 4: Besides Debye-Hückel theory, if there is any other numerical method related to electrolyte dependence of E . Novelty could be further discussed at the beginning.

Reviewer #2 (Remarks to the Author):

This paper quantitatively investigated the electrode potential based on the MD simulations which would be beneficial to help understand the limitations of the classic Debye-Hückel theory. It interpreted the hitherto-unexplained huge experimental shift for the lithium metal electrode through the “liquid Madelung potential” (ELM). This article provided guidance for future researchers to gain a deeper understanding of the fundamental principles of electrochemistry and to predict electrode potentials under high Li^+ concentration conditions. However, there are still many questions and areas for improvement regarding this paper:

1. The content studied in this article is highly fundamental and theoretical, which may pose some difficulties for readers. It is suggested that the authors provide a more detailed introduction to the electrode potential E in the background: How it affects electrochemical systems? What impact would bring if we overlook the potential upshift in daily electrochemical studies? The author should give some more evidence highlighting the practical significance of accurately quantify the potential shift in the electrode.
2. The Debye-Huckel theory leads to calculation inaccuracies when the infinite dilution assumption is not valid. Has this issue been previously investigated? If so, the background section should provide a comprehensive summary in this area. It is necessary to give the advantages of the theoretical model in this paper compared with the previous theoretical model.
3. From Fig 3, it can be observed that there is still a certain shift between the author's theoretically calculated potential curves and the actual experimental curves, especially in Fig 3(c), where the shift between the calculated curve and the experimental curve is almost

identical to that of the curve based on the Nernst equation and the experimental curve. Does this imply that the methodology utilized in this article remains inadequate in accurately forecasting electrode potential?

4. The author examined the upshift curves based on Liquid Madelung potential in only three different FSI-based electrolytes. However, the accuracy predicted from the results appears to be inconsistent. Do these three results alone suffice to demonstrate the universality of the method described in the paper?

5. What is the implementation detail when applying particle mesh Ewald method in calculating the local potential of Li^+ , i.e. the liquid Madelung (Ewald) potential ? Was this accomplished by some mature package or by a customized script ? The authors should give some more details about this.

6. The author mentioned that “ELM is a converged value with excellent quantitative accuracy that is obtained by taking into account the Coulombic interactions of all constituent atoms up to infinite distances” in Page 10. The finite size of the simulation box seems to contradict the “infinite distances” notion, please justify this.

7. The coordination of Li^+ was characterized by both Raman spectroscopy and molecular dynamics simulation targeting at the bulk electrolyte. However, the primary correlation here should be the electrolyte adjacent to the electrode with respect to the electrode. Why not directly probe the coordination near the electrolyte-electrode interface?

8. Similar with the aforementioned point, the authors seem to ignore the influence of the solid electrolyte interface (SEI). What role does SEI play in affecting the potential shift in ELM or Eelectrode, as well as the correlation between ELM and electrode.

Reviewer #3 (Remarks to the Author):

The manuscript reported a new method based on liquid Madelung potential concept to accurately express electrode potential in the concentrated regime, based on which the physical meaning and mechanism governing potential shift from diluted to concentrated regime. The manuscript has issues in novelty and computation accuracy as follows:

(1) The concept of “Madelung potential” has already been applied in similar systems such as lithium metal electrode (DOI: 10.21203/rs.3.rs-1830373/v1) and ionic liquid electrolyte (J. Chem. Phys., 2008, 129, 224507; J. Electron Spectrosc. Relat. Phenom., 2009, 174, 110-115). Therefore, the novelty of this work compared with previous studies on secondary batteries and ionic liquid electrolyte should be highlighted.

(2) This work performed molecular dynamic simulations on pure electrolyte solution to estimate Madelung potential. However, electrolyte ions that participate in electrochemical reaction are usually located near electrode material. Will the electrode material in practical electrochemical system influence the calculation of Madelung potential?

(3) The accuracy of Madelung potential estimation is highly related to the calculation of atomic charge. For electrolyte, the valued of atomic charge may be influenced by surrounding solvent environment. Did the author consider related effects such as implicit solvent model for atomic charge calculation?

(4) The value of atomic charge is dependent on the used functional and basis set. In this regard, the reasons for selecting the used functional and basis set in this work should be described in the manuscript.

Answer to Reviewer #1:

We would like to thank the reviewer for the useful comments to improve our manuscript. In accordance with the comments, we have revised the manuscript.

Comment 1

The author has well introduced the explicit method to interpret the huge potential shift beyond Debye-Hückel's theory. In this work, Li^+ is fully discussed, while I'm curious if this method can also extend to other multivalent metal-ion batteries. If the solvation structure is unclear or complicated if this method still works.

Answer

The original concept, liquid Madelung potential (E_{LM}), is valid for other alkali metals as well as for multivalent metal ions. To support this assertion, we have made additional experimental and computational studies on $\text{Zn}(\text{TFSI})_2/\text{PC}$ solutions, and revised the manuscript accordingly.

Authors' modification to the Manuscript or Supplementary Information:

(lines 8-11, page 9 in the revised manuscript)

Noteworthy is that the concept of E_{LM} is valid for other alkali metals (to be reported elsewhere) as well as for multivalent metal ions, as typically demonstrated for zinc bis(trifluoromethanesulfonyl)imide ($\text{Zn}(\text{TFSI})_2/\text{PC}$) electrolytes in Fig. S3.

(page 6 in the revised supplementary information)

Fig. S3: Experimental potential shifts ($\Delta E_{\text{Zn}/\text{Zn}^{2+}}$ (Exp.)) and the calculated values from liquid Madelung potential ($\Delta E_{LM}/F$) in $\text{Zn}(\text{TFSI})_2/\text{PC}$ electrolytes as a function of salt concentration $m_{\text{Zn}^{2+}}$ with reference to 0.25 mol kg^{-1} . The black dashed line represents the potential shift based on the ideal Nernst equation, where $\gamma_{\text{Zn}^{2+}}$ is assumed to be unity.

Comment 2

In terms of accuracy, the author mentioned “excellent quantitative accuracy”. A detailed discussion regarding accuracy should be addressed. If there is any assumption before using this method. What are these conditions?

Answer

The phrase “excellent quantitative accuracy” represents that the Ewald method takes into account all electrostatic interactions around Li^+ to infinite distance, to distinguish this work from other conventional extended-Debye-Hückel theories (e.g. single ion model or Pitzer equation). To avoid misunderstanding and any confusions, we have deleted the word “with excellent quantitative accuracy” in the revised manuscript.

Authors’ modification to the Manuscript or Supplementary Information:

(lines 13-15, page 11 in the revised manuscript)

Note that E_{LM} is a converged value that is obtained by taking into account the Coulombic interactions of all constituent atoms up to infinite distances.

(Fig. 5g caption, page 10 in the revised manuscript)

It should be noted that E_{LM} is a converged value that is obtained by taking into account the electrostatic interactions of all constituent atoms up to infinite distances.

Comment 3

In this work a high concentration was used as an experimental result, does this also work for super high concentrated electrolytes? Such as water-in-salt electrolytes or deep eutectic electrolytes.

Answer

The solvent-to-salt ratio of LiFSI/EC and LiFSI/PC electrolytes in this work is ca. 1:1.5, which is already in the range of super high concentrated electrolyte. As replied for Comment 1 above, the liquid Madelung potential offers explicit treatment of overall Coulombic interactions in the electrolyte, serving as the general descriptor of electrode potential in any types of electrolyte.

Comment 4

Besides Debye-Hückel theory, if there is any other numerical method related to electrolyte dependence of E. Novelty could be further discussed at the beginning.

Answer

Following the reviewer's opinion, we have modified our manuscript.

Authors' modification to the Manuscript or Supplementary Information:

(lines 21-24, page 3 in the revised manuscript)

Tremendous efforts have been devoted to extend the Debye-Hückel theory to practical concentration region, however, explicit treatment of the long-range Coulombic interaction in highly fluctuating many-body liquid system has long been unrealistic,^{4,5} until the recent advances and maturity in molecular dynamics (MD) simulations.

Answer to Reviewer #2:

We would like to thank the reviewer for the constructive remarks to improve our manuscript. In accordance with the comments, we have revised our manuscript.

Comment 1

The content studied in this article is highly fundamental and theoretical, which may pose some difficulties for readers. It is suggested that the authors provide a more detailed introduction to the electrode potential E in the background: How it affects electrochemical systems? What impact would bring if we overlook the potential upshift in daily electrochemical studies? The author should give some more evidence highlighting the practical significance of accurately quantify the potential shift in the electrode.

Answer

Following the reviewer's opinion, we have modified our manuscript.

Authors' modification to the Manuscript or Supplementary Information:

(lines 12-18, page 3 in the revised manuscript)

Many of the recent studies have revealed that long-lasting challenges in next-generation batteries can be readily addressed through the electrolyte design based on the strategic control of the electrode potential. For instance, a highly reversible Li plating/stripping was achieved by increasing the electrode potential of Li metal ($E_{\text{Li}/\text{Li}^+}$), thereby reducing the thermodynamic driving force for electrolyte reduction.¹ Furthermore, exceptionally stable $\text{SiO}_x/\text{LiNi}_{0.5}\text{Mn}_{1.5}\text{O}_4$ batteries have been developed by optimizing the overall potential diagram of the electrodes and electrolytes.²

Comment 2

The Debye-Huckel theory leads to calculation inaccuracies when the infinite dilution assumption is not valid. Has this issue been previously investigated? If so, the background section should provide a comprehensive summary in this area. It is necessary to give the advantages of the theoretical model in this paper compared with the previous theoretical model.

Answer

Following the reviewer's opinion, we have modified our manuscript.

Authors' modification to the Manuscript or Supplementary Information:

(lines 21-24, page 3 in the revised manuscript)

Tremendous efforts have been devoted to extend the Debye-Hückel theory to practical concentration

region, however, explicit treatment of the long-range Coulombic interaction in highly fluctuating many-body liquid system has long been unrealistic,^{4,5} until the recent advances and maturity in molecular dynamics (MD) simulations.

Comment 3

From Fig 3, it can be observed that there is still a certain shift between the author's theoretically calculated potential curves and the actual experimental curves, especially in Fig 3(c), where the shift between the calculated curve and the experimental curve is almost identical to that of the curve based on the Nernst equation and the experimental curve. Does this imply that the methodology utilized in this article remains inadequate in accurately forecasting electrode potential?

Answer

Since the liquid Madelung potential (E_{LM}) contributes to the enthalpy term, the difference between $\Delta E_{Li/Li+}$ and $\Delta E_{LM}/F$ may partly originate from the entropy term, which is challenging to calculate. Especially, concentrated electrolytes with a sulfolane (SL) solvent (Fig. 3c) are known to exhibit a rapid Li^+ hopping transport mechanism due to their high Li^+ -ligand exchange mobility (*J. Phys. Chem. B*, **122**, 10736 (2018)). This unique property of SL might increase the entropy in the system, causing the deviation from the calculation results. Based on the reviewer's comments, we emphasized the importance of considering the entropy term.

Authors' modification to the Manuscript or Supplementary Information:

(lines 17-18, page 3 in the revised supplementary information)

However, the entropy term and bonding effect are not considered in the present simple model, requiring further sophistication in modeling to achieve higher accuracy.^{1,2}

(lines 1-7, page 16 in the revised supplementary information)

Supplementary References

1. Dokko, K.; Watanabe, D.; Ugata, Y.; Thomas, M. L.; Tsuzuki, S.; Shinoda, W.; Hashimoto, K.; Ueno, K.; Umebayashi, Y.; Watanabe, M. Direct Evidence for Li Ion Hopping Conduction in Highly Concentrated Sulfolane-Based Liquid Electrolytes. *J. Phys. Chem. B* **122**, 10736-10745 (2018).
2. Xu, J. High-entropy electrolytes in boosting battery performance. *Mater. Futures* **2**, 047501.

Comment 4

The author examined the upshift curves based on Liquid Madelung potential in only three different FSI-based electrolytes. However, the accuracy predicted from the results appears to be inconsistent. Do these three results alone suffice to demonstrate the universality of the method described in the paper?

Answer

The original concept, liquid Madelung potential (E_{LM}), is valid for other alkali metals as well as for multivalent metal ions. To support this assertion, we have made additional experimental and computational studies on $Zn(TFSI)_2/PC$ solutions, and revised the manuscript accordingly.

Authors' modification to the Manuscript or Supplementary Information:

(lines 9-11, page 9 in the revised manuscript)

Noteworthy is that the concept of E_{LM} is valid for other alkali metals (to be reported elsewhere) as well as for multivalent metal ions, as typically demonstrated for zinc bis(trifluoromethanesulfonyl)imide ($Zn(TFSI)_2/PC$) electrolytes in Fig. S3.

(page 6 in the revised supplementary information)

Fig. S3: Experimental potential shifts ($\Delta E_{Zn/Zn^{2+}}$ (Exp.)) and the calculated ones from liquid Madelung potential ($\Delta E_{LM}/F$) in $Zn(TFSI)_2/PC$ electrolytes as a function of salt concentration $m_{Zn^{2+}}$ with reference to 0.25 mol kg^{-1} . The black dashed line represents the potential shift based on the ideal Nernst equation, where $\gamma_{Zn^{2+}}$ is assumed to be unity.

Comment 5

What is the implementation detail when applying particle mesh Ewald method in calculating the local potential of Li⁺, i.e. the liquid Madelung (Ewald) potential? Was this accomplished by some mature package or by a customized script? The authors should give some more details about this.

Answer

We evaluated the liquid Madelung potential by modifying the source code of the MD calculation program (AMBER software package). We have added the details to the revised manuscript.

Authors' modification to the Manuscript or Supplementary Information:

(lines 15-17, page 7 in the revised manuscript)

Note that E_{LM} is calculated based on the particle mesh Ewald method (see S1 in the Supplementary Information for more details).¹⁴

(lines 1-13, page 2 in the revised supplementary information)

S1. Liquid Madelung potential calculation

In this study, E_{LM} is obtained numerically by calculating the Coulombic interaction energies between the target Li⁺ ion and other solvents/ions during the MD simulations under periodic boundary conditions and subsequently averaging these values over time for all Li⁺ ions as follows:

$$E_{LM} = \frac{1}{N_{Li^+}} \left\langle \sum_i (E^{tot} - E_i^{extracted} - E^{Li^+}) \right\rangle \quad [S1]$$

where E^{tot} is the total Coulombic energy of the system, $E_i^{extracted}$ is the Coulombic energy of the system with the i -th Li⁺ extracted, E^{Li^+} is the Coulombic energy of the system with one Li⁺, and N_{Li^+} is the number of Li⁺. Note that the Coulombic energies of the charge excess systems ($E^{extracted}$ and E^{Li^+}) under periodic boundary conditions include the numerical errors due to the background charges to neutralize the system, depending on the simulation cell size. To mitigate this unavoidable effect, the simulation cell size was kept almost the same for each electrolyte.

Comment 6

The author mentioned that “ELM is a converged value with excellent quantitative accuracy that is obtained by taking into account the Coulombic interactions of all constituent atoms up to infinite distances” in Page 10. The finite size of the simulation box seems to contradict the “infinite distances” notion, please justify this.

Answer

In MD simulations, employing the periodic boundary conditions is an established method to represent electrolyte solutions, allowing the calculation of Coulombic interactions over infinite distances. More importantly, the electrostatic interactions around Li^+ are shielded within ca. 20 Å, corresponding to the solvation shell size of Li^+ (see Fig. S2). Therefore, the box size in our current MD simulations (~70 Å per side) is large enough, considering the much smaller solvation shell size (i.e., 20 Å).

Comment 7

The coordination of Li^+ was characterized by both Raman spectroscopy and molecular dynamics simulation targeting at the bulk electrolyte. However, the primary correlation here should be the electrolyte adjacent to the electrode with respect to the electrode. Why not directly probe the coordination near the electrolyte-electrode interface?

Answer

Regardless of the interface structure, the electrode potential is given as the difference of inner potential, Φ_{Li} , in the bulk electrode and inner potential, $\Phi_{\text{Li}^+}^{\text{SSIP}}$ or $\Phi_{\text{Li}^+}^{\text{AGG}}$, in the bulk electrolyte solution (*Electrochemistry*, **90**, 102001 (2022)), as illustrated in the diagram below.

Fig. R1: Schematic diagram showing the relationship between inner potential and electrode potential shift.

As Φ_{Li} is independent of the electrolyte used, $\Delta E_{\text{Li}/\text{Li}^+}$, the shift of the electrode potential between the low and high concentration electrolytes, is shown as follows.

$$\Delta E_{\text{Li}/\text{Li}^+} = E_{\text{Li}/\text{Li}^+}^{\text{AGG}} - E_{\text{Li}/\text{Li}^+}^{\text{SSIP}} = (\Phi_{\text{Li}} - \Phi_{\text{Li}^+}^{\text{AGG}}) - (\Phi_{\text{Li}} - \Phi_{\text{Li}^+}^{\text{SSIP}}) = \Phi_{\text{Li}^+}^{\text{SSIP}} - \Phi_{\text{Li}^+}^{\text{AGG}}$$

Based on this relationship, we demonstrate that major contribution to $\Phi_{\text{Li}^+}^{\text{SSIP}} - \Phi_{\text{Li}^+}^{\text{AGG}}$ is the Coulombic energy penalty.

Comment 8

Similar with the aforementioned point, the authors seem to ignore the influence of the solid electrolyte interface (SEI). What role does SEI play in affecting the potential shift in ELM or Eelectrode, as well as the correlation between ELM and electrode.

Answer

SEI does not influence the electrode potential of lithium $E_{\text{Li}/\text{Li}^+}$, because $E_{\text{Li}/\text{Li}^+}$ is independent of the chemical potential of Li^+ in the SEI, as proved in our recent paper (*Nature Energy*, **7**, 1217 (2022)). Specifically, $E_{\text{Li}/\text{Li}^+}$ was measured as an electromotive force of the cell consisting of six phases (I, II, III, IV, V and VI) in this paper. T(I) |Pt(II)|Fc, Fc^+ , Li^+ (III)| Li^+ , SEI(IV)|Li(V)|T(VI), where T(I) and T(VI) denote both metal terminals of the cell. On this basis, the $E_{\text{Li}/\text{Li}^+}$ is derived as follows:

$$\begin{aligned} E_{\text{Li}/\text{Li}^+} &= \phi^{\text{VI}} - \phi^{\text{I}} = (\phi^{\text{VI}} - \phi^{\text{IV}}) + (\phi^{\text{IV}} - \phi^{\text{III}}) + (\phi^{\text{III}} - \phi^{\text{I}}) \\ &= \frac{\mu_{\text{Li}^+}^{\text{IV}} + \mu_{\text{e}}^{\text{VI}} - \mu_{\text{Li}}^{\text{V}}}{F} + \frac{\mu_{\text{Li}^+}^{\text{III}} - \mu_{\text{Li}^+}^{\text{IV}}}{F} + \frac{\mu_{\text{Fc}}^{\text{III}} - \mu_{\text{Fc}^+}^{\text{III}} - \mu_{\text{e}}^{\text{I}}}{F} = \frac{\mu_{\text{Li}^+}^{\text{III}} + \mu_{\text{e}}^{\text{VI}} - \mu_{\text{Li}}^{\text{V}}}{F} + \frac{\mu_{\text{Fc}}^{\text{III}} - \mu_{\text{Fc}^+}^{\text{III}} - \mu_{\text{e}}^{\text{I}}}{F} \end{aligned}$$

where ϕ , μ , and F denote the inner potential of each phase (I, II, III, IV, V, VI), the chemical potential of each species in each phase, and Faraday constant, respectively. According to IUPAC, the redox potential of Fc/Fc^+ , $\phi^{\text{I}} - \phi^{\text{III}} = -(\mu_{\text{Fc}}^{\text{III}} - \mu_{\text{Fc}^+}^{\text{III}} - \mu_{\text{e}}^{\text{I}})/F$, is assumed to be constant independent of electrolyte compositions. Besides, $\mu_{\text{e}}^{\text{VI}}$ and $\mu_{\text{Li}}^{\text{V}}$ are also independent of the electrolyte used. On this basis, the $E_{\text{Li}/\text{Li}^+}$ is shown as follows.

$$E_{\text{Li}/\text{Li}^+} = \frac{\mu_{\text{Li}^+}^{\text{III}}}{F} + \text{const.}$$

Thus, the observed variations in $E_{\text{Li}/\text{Li}^+}$ was derived from the different chemical potential of Li^+ in the electrolytes ($\mu_{\text{Li}^+}^{\text{III}}$), not the chemical potential of Li^+ in the SEI ($\mu_{\text{Li}^+}^{\text{IV}}$).

Similarly, E_{LM} in the SEI does not influence the electrode potential shift. Indeed, as shown in Fig. 2d, the potential shifts for Li metal, LTO, and LFP are completely identical. To emphasize this point, we added the following sentences.

Authors' modification to the Manuscript or Supplementary Information:

(lines 6-7, page 6 in the revised manuscript)

The identical potential upshifts also indicate that a passivation film on the electrodes has no impact on

E .¹

Answer to Reviewer #3:

We would like to thank the reviewer for the useful comments to improve our manuscript. In accordance with the comments, we have revised our manuscript.

Comment 1

The concept of “Madelung potential” has already been applied in similar systems such as lithium metal electrode (DOI: 10.21203/rs.3.rs-1830373/v1) and ionic liquid electrolyte (*J. Chem. Phys.*, 2008, 129, 224507; *J. Electron Spectrosc. Relat. Phenom.*, 2009, 174, 110-115). Therefore, the novelty of this work compared with previous studies on secondary batteries and ionic liquid electrolyte should be highlighted.

Answer

This manuscript is the revised version of the preprint (DOI: 10.21203/rs.3.rs-1830373/v1) by ourselves, which has not been published elsewhere. According to the reviewer’s comments, we also highlighted the discussion on the novelty of our work compared with previous studies for ionic liquid electrolytes as shown below.

Authors’ modification to the Manuscript or Supplementary Information:

(lines 11-13, page 9 in the revised manuscript)

To emphasize, the liquid Madelung potential has never been calculated directly according to Eq. [2], though the term was used for conceptual interpretation of the experimental spectra measured for ionic liquids,¹⁷⁻²⁰

(line 23, page 16 – line 2, page17 in the revised manuscript)

19. Kanai, K.; Nishi, T.; Iwahashi, T.; Ouchi, Y.; Seki, K.; Harada, Y.; Shin, S. Anomalous electronic structure of ionic liquids determined by soft x-ray emission spectroscopy: contributions from the cations and anions to the occupied electronic structure. *J. Chem. Phys.* **129**, 224507 (2008).

20. Kanai, K.; Nishi, T.; Iwahashi, T.; Ouchi, Y.; Seki, K.; Harada, Y.; Shin, S.; Electronic structures of imidazolium-based ionic liquids. *J. Electron Spectrosc. Relat. Phenom.* **174**, 110–115 (2009).

Comment 2

This work performed molecular dynamic simulations on pure electrolyte solution to estimate Madelung potential. However, electrolyte ions that participate in electrochemical reaction are usually located near electrode material. Will the electrode material in practical electrochemical system influence the calculation of Madelung potential?

Answer

Regardless of the interface structure, the electrode potential is given as the difference of inner potential, Φ_{Li} , in the bulk electrode and inner potential, $\Phi_{\text{Li}^+}^{\text{SSIP}}$ or $\Phi_{\text{Li}^+}^{\text{AGG}}$, in the bulk electrolyte solution (*Electrochemistry*, **90**, 102001 (2022)), as illustrated in the diagram below.

Fig. R1: Schematic diagram showing the relationship between inner potential and electrode potential shift.

As Φ_{Li} is independent of the electrolyte used, $\Delta E_{\text{Li/Li}^+}$, the shift of the electrode potential between the low and high concentration electrolytes, is shown as follows.

$$\Delta E_{\text{Li/Li}^+} = E_{\text{Li/Li}^+}^{\text{AGG}} - E_{\text{Li/Li}^+}^{\text{SSIP}} = (\Phi_{\text{Li}} - \Phi_{\text{Li}^+}^{\text{AGG}}) - (\Phi_{\text{Li}} - \Phi_{\text{Li}^+}^{\text{SSIP}}) = \Phi_{\text{Li}^+}^{\text{SSIP}} - \Phi_{\text{Li}^+}^{\text{AGG}}$$

Based on this relationship, we demonstrate that major contribution to $\Phi_{\text{Li}^+}^{\text{SSIP}} - \Phi_{\text{Li}^+}^{\text{AGG}}$ is the Coulombic energy penalty. In fact, as shown in Fig. 2d, the potential shifts for Li metal, LTO, and LFP are completely identical, clearly indicating that they are independent of the electrode material.

Comment 3

The accuracy of Madelung potential estimation is highly related to the calculation of atomic charge. For electrolyte, the value of atomic charge may be influenced by surrounding solvent environment. Did the author consider related effects such as implicit solvent model for atomic charge calculation?

Answer

We used atomic charges based on the restrained electrostatic potential (RESP), which is widely used to describe the intermolecular interactions in liquid-phase MD simulations. Also, a use of commonly accepted empirical scaling factor of 0.8 (refs. 29-33) to adjust ion charge was enough to reproduce the experimental solution structures of electrolyte used in this study. Of course, we agree with the reviewer's opinion and now try to incorporate the polarization effect for more general feasibility.

Comment 4

The value of atomic charge is dependent on the used functional and basis set. In this regard, the reasons for selecting the used functional and basis set in this work should be described in the manuscript.

Answer

Based on the reviewer's opinion, we have evaluated the Madelung potential using different RESP charge calculation level under the identical ionic charge scaling factor (0.8).

Authors' modification to the Manuscript or Supplementary Information:

(lines 15-17, page 14 in the revised manuscript)

The potentials were compared using different calculation levels, HF and M06-2x, where all methods show reasonable agreement with the experimental potential shift (Fig. S10).

(page 13 in the revised supplementary information)

Fig. S10: Experimental data for $\Delta E_{Li/Li^+}$ (black circles) and calculated potential shifts of liquid Madelung potentials ($\Delta E_{LM}/F$) using RESP charges at various calculation levels (red circles: B3LYP/cc-pvdz (this work), blue circles: HF/cc-pvdz, green circles: M06-2x/cc-pvdz) in LiFSI/EC electrolytes as a function of salt concentration m_{Li^+} with reference to 0.8 mol kg⁻¹.

REVIEWER COMMENTS

Reviewer #2 (Remarks to the Author):

The manuscript has been improved, I still have some comments as below:

1. According to Eq. (S1) in the response letter, it seems that E_{tot} , $E_{extracted}$, and E_{Li+} are separately calculated, where $E_{extracted}$, and E_{Li+} contain the numerical errors due to the background charges to counterbalance the excess charge. If this is the case, the author should give some more specific information about the computation settings, especially for the system pertaining to $E_{extracted}$, and E_{Li+} , as well as the consistency between each calculation from a computational standpoint.
2. In our last comment "The coordination of $Li+$ was characterized by both Raman spectroscopy and molecular dynamics simulation targeting at the bulk electrolyte. However, the primary correlation here should be the electrolyte adjacent to the electrode with respect to the electrode. Why not directly probe the coordination near the electrolyte-electrode interface?", we suspected that the coordination of $Li+$ might be altered as it approached the electrolyte-electrode interface. Since the author tended to relate the variation of ELM to $Li+$ coordination and its electron density distribution, the interfacial $Li+$ coordination should be probed instead of the bulk $Li+$ coordination. The relation between $\Delta E_{Li/Li+}$ and $\Delta\phi$ was not the question here. Furthermore, the derived relation between $\Delta E_{Li/Li+}$ and $\Delta\phi$ seems cannot hold in the author's response, because $\Delta E_{Li/Li+} > 0$ and $\phi_{SSIP} - \phi_{AGG} < 0$ (ϕ_{SSIP} is more negative).
3. In the response to justifying that SEI does not influence the electrode potential $E_{Li/Li+}$, the author expressed $E_{Li/Li+}$ in terms of the chemical potential in each phase. However, the whole SEI viewed as one phase should be questioned here since SEI is not physicochemically uniform. The author should justify their assumption that SEI could be regarded as one single phase. As for the experimental observation that the electrode potential almost had identical shift when different electrolytes were used, has any other group achieved similar results?

Reviewer #3 (Remarks to the Author):

The responses and revised manuscript have resolved most problems proposed by the reviews. I still have a following small question to discuss with the author:

It can be seen from Fig. S10 that as to the salt with high concentration, the calculation level for RESP charge does influence the predicted value of potential shift as well as the prediction accuracy. Can the author give suggestion on the selection of suitable calculation level for RESP charge from the prospect of accuracy and calculation efficiency.

Answer to Reviewer #2:

We would like to thank the reviewer for the constructive remarks to improve our manuscript. In accordance with the comments, we have revised our manuscript.

Comment 1

According to Eq. (S1) in the response letter, it seems that E_{tot} , $E_{extracted}$, and E_{Li+} are separately calculated, where $E_{extracted}$, and E_{Li+} contain the numerical errors due to the background charges to counterbalance the excess charge. If this is the case, the author should give some more specific information about the computation settings, especially for the system pertaining to $E_{extracted}$, and E_{Li+} , as well as the consistency between each calculation from a computational standpoint.

Answer

The errors are negligible, because the calculated potential shifts with background charges (BC) showed almost identical values with those obtained by directly summing the Coulombic interactions (without BC) among Li^+ and other solvents/ions up to the electrostatically shielded distance (i.e., 30 Å) (Fig.S11).

(lines 13-16, page 2 in the revised supplementary information)

The errors are indeed negligible, because the calculated potential shifts with BC showed almost identical values with those obtained by directly summing the Coulombic interactions (without BC) among Li^+ and other solvents/ions up to the electrostatically shielded distance (i.e., 30 Å) (Fig. S11).

(page 14 in the revised supplementary information)

Fig. S11: Calculated potential shifts of liquid Madelung potentials ($\Delta E_{LM}/F$) using two different methods (blue circles: with background charges (BC) (this work), red circles: without BC) in LiFSI/EC

electrolytes as a function of salt concentration m_{Li^+} with reference to 0.8 mol kg^{-1} . Differences with and without BC are also indicated by black circles. The potential shifts without BC are obtained by directly summing the Coulomb interactions among Li^+ and other solvents/ions up to the electrostatically shielded distance (i.e., 30 \AA).

Comment 2

In our last comment “The coordination of Li^+ was characterized by both Raman spectroscopy and molecular dynamics simulation targeting at the bulk electrolyte. However, the primary correlation here should be the electrolyte adjacent to the electrode with respect to the electrode. Why not directly probe the coordination near the electrolyte-electrode interface?”, we suspected that the coordination of Li^+ might be altered as it approached the electrolyte-electrode interface. Since the author tended to relate the variation of ELM to Li^+ coordination and its electron density distribution, the interfacial Li^+ coordination should be probed instead of the bulk Li^+ coordination. The relation between $\Delta E_{\text{Li}/\text{Li}^+}$ and $\Delta\phi$ was not the question here. Furthermore, the derived relation between $\Delta E_{\text{Li}/\text{Li}^+}$ and $\Delta\phi$ seems cannot hold in the author’s response, because $\Delta E_{\text{Li}/\text{Li}^+} > 0$ and $\phi_{\text{SSIP}} - \phi_{\text{AGG}} < 0$ (ϕ_{SSIP} is more negative).

Answer

The Li electrode potential ($E_{\text{Li}/\text{Li}^+}$) is given by the difference in electromotive force between the working and reference electrodes and therefore determined solely by the chemical potential of Li^+ in the bulk electrolyte, because the contribution of the chemical potential of Li^+ at the interface is canceled out. Thus, direct probing of the electrode surface is not necessary to obtain $E_{\text{Li}/\text{Li}^+}$.

On the second inquiry, the discrepancy simply comes from our mistake in formula/figure expression. We appreciate the reviewer for notifying. In the corrected formula/figure below, $\Delta E_{\text{Li}/\text{Li}^+}$ becomes positive ($\phi_{\text{Li}^+}^{\text{AGG}} - \phi_{\text{Li}^+}^{\text{SSIP}} > 0$).

Corrected Figure: Schematic diagram showing the relationship between inner potential and electrode potential shift.

Corrected formula:

$$\Delta E_{\text{Li}/\text{Li}^+} = E_{\text{Li}/\text{Li}^+}^{\text{SSIP}} - E_{\text{Li}/\text{Li}^+}^{\text{AGG}} = (\phi_{\text{Li}} - \phi_{\text{Li}^+}^{\text{SSIP}}) - (\phi_{\text{Li}} - \phi_{\text{Li}^+}^{\text{AGG}}) = \phi_{\text{Li}^+}^{\text{AGG}} - \phi_{\text{Li}^+}^{\text{SSIP}}$$

Comment 3

In the response to justifying that SEI does not influence the electrode potential $E_{\text{Li}/\text{Li}^+}$, the author expressed $E_{\text{Li}/\text{Li}^+}$ in terms of the chemical potential in each phase. However, the whole SEI viewed as one phase should be questioned here since SEI is not physicochemically uniform. The author should justify their assumption that SEI could be regarded as one single phase. As for the experimental observation that the electrode potential almost had identical shift when different electrolytes were used, has any other group achieved similar results?

Answer

Our previous reply holds true even if the SEI consists of multiple phases. If the SEI is divided into n parts (SEI(1), ..., SEI(n)), $E_{\text{Li}/\text{Li}^+}$ is measured as an electromotive force of the cell consisting of $n + 5$ phases (T(I)|Pt(II)|Fc, Fc⁺, Li⁺(III)|Li⁺, SEI(n)|...|Li⁺, SEI(1)|Li(IV)|T(V)). On this basis, the $E_{\text{Li}/\text{Li}^+}$ is derived as follows:

$$\begin{aligned} E_{\text{Li}/\text{Li}^+} &= \phi^{\text{V}} - \phi^{\text{I}} = (\phi^{\text{V}} - \phi^{\text{SEI}(1)}) + \sum_{i=2}^n (\phi^{\text{SEI}(i-1)} - \phi^{\text{SEI}(i)}) + (\phi^{\text{III}} - \phi^{\text{I}}) \\ &= \frac{\mu_{\text{Li}^+}^{\text{SEI}(1)} + \mu_{\text{e}}^{\text{V}} - \mu_{\text{Li}}^{\text{IV}}}{F} + \sum_{i=2}^n \left(\frac{\mu_{\text{Li}^+}^{\text{SEI}(i)} - \mu_{\text{Li}^+}^{\text{SEI}(i-1)}}{F} \right) + \frac{\mu_{\text{Li}^+}^{\text{III}} - \mu_{\text{Li}^+}^{\text{SEI}(n)}}{F} + \frac{\mu_{\text{Fc}}^{\text{III}} - \mu_{\text{Fc}^+}^{\text{III}} - \mu_{\text{e}}^{\text{I}}}{F} \\ &= \frac{\mu_{\text{Li}^+}^{\text{III}} + \mu_{\text{e}}^{\text{V}} - \mu_{\text{Li}}^{\text{IV}}}{F} + \frac{\mu_{\text{Fc}}^{\text{III}} - \mu_{\text{Fc}^+}^{\text{III}} - \mu_{\text{e}}^{\text{I}}}{F} = \frac{\mu_{\text{Li}^+}^{\text{III}}}{F} + \text{const.} \end{aligned}$$

Thus, the SEI structure does not influence $E_{\text{Li}/\text{Li}^+}$.

On the identical electrode potential shift with different electrolyte, it is the definite outcome of thermodynamic requirement. A straightforward expression of this is Figure 2. We emphasize again that the potentials of all Li⁺-related reactions (e.g., Li-metal anode, Li₄Ti₅O₁₂ anode, and LiFePO₄ cathode) depend on the change in the Gibbs free energy of Li⁺ in the electrolyte, and hence the shifts occur simultaneously in identical magnitudes. This is why the battery voltage (a two-electrode cell) does not change regardless of electrolyte composition even with a significant potential shift in both the anode and cathode. The potential shift of each electrode can be observed only when it is measured relative to the standard electrode such as ferrocene (Fc/Fc⁺), of which potential remains independent of the electrolyte.

Answer to Reviewer #3:

We would like to thank the reviewer for the useful comments to improve our manuscript. In accordance with the comments, we have revised our manuscript.

Comment 1

It can be seen from Fig. S10 that as to the salt with high concentration, the calculation level for RESP charge does influence the predicted value of potential shift as well as the prediction accuracy. Can the author give suggestion on the selection of suitable calculation level for RESP charge from the prospect of accuracy and calculation efficiency?

Answer

Although the present calculation level incorporating electronic correlations gave reasonable agreement with experiment, we are now introducing the polarization effects into MD simulations that can further improve the calculation accuracy and mitigate the dependence on the calculation level, the details of which will be reported elsewhere.

REVIEWERS' COMMENTS

Reviewer #2 (Remarks to the Author):

The authors have addressed the reviewer's comments and the manuscript can be accepted now.

Reviewer #3 (Remarks to the Author):

The question has been answered.